# Percutaneous Endoscopic Gastrostomy and Nutritional Interventions by the Pediatric Nutritional Support Team Improve the Nutritional Status of Neurologically Impaired Children

**DOI:** 10.3390/jcm9103295

**Published:** 2020-10-14

**Authors:** Chae-ri Suh, Wonkyung Kim, Baik-Lin Eun, Jung Ok Shim

**Affiliations:** 1Department of Pediatrics, Korea University College of Medicine, Korea University Guro Hospital 1, Seoul 08308, Korea; ch2307@naver.com (C.-r.S.); bleun@korea.ac.kr (B.-L.E.); 2Pediatric Nutritional Support Team, Korea University Guro Hospital, Seoul 08308, Korea; osim17@naver.com

**Keywords:** child, gastrostomy, growth, malnutrition, nutritional support

## Abstract

Aim: To evaluate the long-term effects of nutritional improvement following percutaneous endoscopic gastrostomy (PEG) tube feeding stratified by previous feeding method and to assess the impact of underlying muscle tone on the outcomes of the nutritional intervention. Methods: Neurologically impaired children who underwent PEG tube insertion and nutritional intervention provided by a pediatric nutritional support team were enrolled. We measured anthropometric variables, laboratory parameters, and nutritional intake at baseline, 6 months after PEG insertion, and the last visit. We evaluated the percent ideal body weight (PIBW), body mass index (BMI)-for-age *z*-score, and percentiles and calculated the ratios of calorie intake compared to required requirement (CIR) and protein intake compared to recommended requirement (PIR). Results: The PIBW and BMI-for-age *z*-score improved during the first 6 months (*p* = 0.003 and *p* = 0.005, respectively). The CIR (*p* = 0.015) and PIR (*p* = 0.004) increased during the study period. The baseline BMI and PIBW of the previous nasogastric tube feeding group were better than those of the oral feeding group (*p* = 0.02 and *p* = 0.03, respectively). The BMI-for-age *z*-score, PIBW, CIR, and PIR improved in the hypertonic group (*p* = 0.03, 0.02, 0.03, and 0.01, respectively). Conclusion: PEG tube feeding and active nutritional intervention improved the nutritional status of neurologically impaired children immediately after PEG insertion. The nutritional requirements might vary by the muscle tonicity.

## 1. Introduction

With advances in medical treatments, the survival of neurologically impaired children has increased. However, along with improved life expectancy, feeding problems are on the rise [1]. Oropharyngeal dysfunction and gastroesophageal reflux are common problems in patients with severe impairments and can result in malnutrition and growth failure [2,3,4,5,6]. Malnutrition also decreases cerebral function; impairs immune function; leads to lean body mass, poor bone health, and micronutrient deficiencies; and diminishes respiratory muscle strength [7,8,9]. Severe malnutrition also leads to increased morbidity and mortality rates; however, 60% of patients are not evaluated for their feeding problems [3]. Thus, these disabilities lead to worsening of the individual’s nutritional status, and subsequently, require specialized nutritional care to correct these problems. Pediatric patients, in particular, need professional and individualized nutritional care to treat complications and ensure proper growth and development.

Percutaneous endoscopic gastrostomy (PEG) has long enabled enteral nutrition in children who are unable to meet their nutritional needs orally [10]. Since many clinicians consider PEG tube feeding as just another feeding route, most previous studies focused on complications of the procedure [11,12] or compared the efficacies of different gastrostomy methods [13].

Nutritional intervention is important, particularly in children with moderate to severe protein-calorie malnutrition. The pediatric nutritional support team (PNST) is a multidisciplinary team consisting of physicians, including pediatric gastroenterologists, neonatologists, surgeons, dieticians, pharmacists, and nurses. A pediatric gastroenterologist coordinates the team as a leader, and the PNST evaluates the nutritional status of every patient who needs enteral or parenteral nutrition and/or any patient who needs special nutritional support under consultation and suggests future nutritional plans to attending doctors. Since there are few studies on the nutritional effect of PEG tube feeding on such patients, we sought to evaluate the ability of PEG tube feeding to improve nutritional status and assessed the efficacy of the regular nutritional intervention by PNST. We compared the nutritional status, including improvement in the anthropometric variables, in the oral feeding group and nasogastric (NG) tube feeding group according to the previous feeding method and muscle tonicity (hypertonic or hypotonic).

## 2. Experimental Section

### 2.1. Study Enrollment

This longitudinal retrospective cohort study was conducted at Korea University Guro Hospital (KUGH) and investigated the nutritional outcomes of PEG tube feeding in neurologically impaired children. The changes in the anthropometric parameters and nutritional status were measured before and after PEG feeding tube insertion. The KUGH Institutional Review Board approved this study (2019GR0354), and the requirement for informed consent was waived, due to the retrospective nature of the study.

Pediatric patients with neurological impairments who underwent PEG tube insertion between 2012 and 2018 at KUGH were eligible to participate. The exclusion criteria were (1) age > 20 years, and (2) a short follow-up period (<6 months). Each patient underwent at least three tests before PEG tube insertion, including a videofluoroscopy swallowing study, 24 h multichannel intraluminal impedance pH monitoring, and esophagogastroduodenoscopy (EGD), to assess the indications for PEG, in which a feeding tube was inserted using a pull-through or push-through technique.

### 2.2. Assessment Schedule

We analyzed the formatted medical records of the PNST at three time points: Immediately before PEG tube insertion (baseline), six months after PEG tube insertion (six months), and at the last hospital visit (last). The PNST evaluated the anthropometric parameters (height and weight) and laboratory tests (hemoglobin, lymphocytes, protein, albumin, total bilirubin, blood urea nitrogen, creatinine, calcium, and phosphorus) to check the nutritional status and general condition of the patients at every visit. The PNST calculated the patients’ nutritional requirements (caloric and protein intake) and recommended an optimal nutritional plan if it was deemed insufficient. The PNST also recorded the following nutritional information: Anthropometric data, previously ingested food types, feeding route (oral, NG tube, or gastrostomy), the volume of food ingested daily, ingested calories and protein, laboratory tests, and recommended type and volume of the formula.

In real practice, the PNST routinely monitored all children in the intensive care unit, those who were receiving enteral nutrition or total parenteral nutrition, and those who were referred by their physicians. The PNST members gathered at least once a week and performed additional monitoring of patients if needed. The pediatric neurologists requested the PEG tube insertion or the PNST suggested tube insertion after the evaluation.

### 2.3. Anthropometric Evaluation and Nutritional Assessment

The ulnar length was measured to predict the patients’ height [14]. Bodyweight was measured using electronic weighing scales (the same scale at each visit). The height and weight were transformed to sex-specific *z*-scores of height-for-age, weight-for-age, body mass index (BMI)-for-age (≥24 months of age), and weight-for-height (<24 months of age) using the 2017 Korea National Growth Charts [15]. The growth curves for cerebral palsy (CP) according to sex and the Gross Motor Function Classification System (GMFCS) level [16] were used to evaluate the percentages of height-for-age, weight-for-age, and BMI-for-age and to calculate the percent of ideal bodyweight (PIBW). IBW was defined using the Moore method [17] and was used to calculate the PIBW or the ratio of IBW to the actual bodyweight.

In addition to gross motor function evaluation, the modified Ashworth scale (MAS) was used to evaluate the patients’ muscle tone [18].

Malnutrition was assessed according to the range of BMI-for-age *z*-score divided by the World Health Organization growth standard as follows: Above 3, obese; above 2, overweight (overnutrition); above −2, normal; between −2 and −3, wasted (malnutrition); and below −3, severely wasted (severe malnutrition) [19]. Waterlow’s classification [20] was used as a reference evaluation for catch-up growth. Waterlow’s classification defined mild malnutrition as 80% to 90% of the PIBW, moderate malnutrition as 70% to 80% of the PIBW, and severe malnutrition as less than 70% of the PIBW. The US Centers for Disease Control definition of weight status category was used to calculate the BMI-for-age percentile as follows [21]: Underweight, below the 5th percentile; normal, between the 5th and 85th percentile; overweight, between the 85th and 95th percentile; and obese, over the 95th percentile. With regard to the weight-for-height *z*-score, the nutritional status of children under 24 months old was assessed as wasted (weight-for-height *z*-score below −2), normal (between −2 and 2), or overweight (over 2) [15].

The PNST evaluated the current nutritional status and calculated the nutritional requirements based on anthropometric variables and laboratory test results. The nutrition level was divided into four stages by severity: Obese, well-nourished (overweight or normal), moderately malnourished (wasted), and severely malnourished (severely wasted) [20]. The nutritional status was evaluated by applying the standard bodyweight of neurologically impaired children. We defined children with a PIBW under 80%, BMI-for-age percentile below the 5th percentile, and BMI-for-age *z*-score below −2 as the malnutrition group. The PNST recommended the amount of feeding formula based on the Schofield equation [22] and the protein requirement, as reported by Williams [23].

### 2.4. Statistical Analysis

The differences in data were examined using the Wilcoxon signed-rank test, and the differences between the oral feeding group and the NG tube feeding group were evaluated using the Mann–Whitney *U* test. *p*-values < 0.05 were considered statistically significant. Analyses were performed using SPSS 24.0 for Windows (SPSS Inc, Chicago, IL, USA).

## 3. Results

### 3.1. Demographic Characteristics

Thirty patients underwent PEG tube insertion during the study period, of whom 12 were excluded for the following reasons: Five children from secondary hospitals who came to our hospital for PEG insertion before returning to the secondary hospital following the procedure and seven children aged > 20 years from the baseline. Table 1 demonstrates the nutritional characteristics of enrolled patients. Among the 18 included patients, 16 were older than 24 months, and two were younger than 24 months. Twelve were males, and six were females. The median age of the patients at baseline was 132.4 (interquartile range (IQR), 43.0–180.7) months, and the median intervention period was 50.4 (IQR, 33.9–71.8) months; the longest intervention period was 71.8 months. All children had intractable epilepsy and were bedridden.

Table 2 shows the characteristics of the enrolled patients, including their neurological problems. In children with hypertonicity, 11 had a MAS score ≥ 2, and 3 had a MAS score of 1 [18]. Sixteen patients underwent PEG tube insertion using the pull-through method, and one underwent insertion using the push-through method. The gastrostomy tube of one patient was re-inserted surgically, due to transverse colonic perforation after the pull-through method. Otherwise, no surgical complications were noted. The baseline laboratory test results were within normal limits (Appendix A).

### 3.2. Changes in Anthropometric Data

According to the BMI-for-age *z*-score of the 2017 Korea National Growth Chart [15], all 16 children aged ≥ 24 months were wasted or severely wasted at the baseline visit. The proportion of severe malnourishment in children aged ≥ 24 months decreased from 56.3% to 38.5% during the study period. One patient was obese by the time of the last visit (Figure 1). Two children aged < 24 months were well-nourished according to the weight-for-height *z*-score, and they maintained their well-nourished status during the study period.

(95% CI: 2.53–29.34, *p* = 0.009), PIBW (95% CI: 0.26–26.92, *p* = 0.041), and height-for-age *z*-score (95% CI: 0.00–1.67, *p* = 0.034) improved significantly at the 6-month and last visits (Table 3). The other anthropometric parameter changes are shown in Table 3.

### 3.3. Changes in Caloric and Protein Intake

Both caloric and protein intake increased significantly after PEG tube insertion (Table 3). The calorie intake increased from 45.4% to 80.0% during the 6-month period (95% CI: 6.21–44.23, *p* = 0.015 compared to the recommended requirement (CIR)) and then to 85.7% at the last visit (95% CI: 14.25–52.52, *p* = 0.016). The protein intake increased from 50.1% to 84.0% (95% CI: 14.66–49.73, *p* = 0.004 compared to the recommended requirement (PIR)) at the 6-month visit and then to 90.0% (95% CI: 24.09–55.73, *p* = 0.002) at the last visit.

### 3.4. Anthropometric Parameters and Nutritional Intake by Previous Feeding Method

The anthropometric data and nutritional assessment data of the oral feeding and NG tube feeding groups were compared. The baseline BMI-for-age *z*-score, BMI-for-age percentile, and PIBW of the NG tube feeding group were significantly better than those of the oral feeding group (*p* = 0.02 for BMI *z*-score, *p* = 0.02 for BMI percentile, and *p* = 0.03 for PIBW) at baseline and 6-month visit. The median BMI *z*-score of the oral feeding group was −4.55 (IQR, −5.97 to −3.85), that of the BMI percentile was 4.64 (IQR, 4.41–5.05), and that of the PIBW was 65.19 (IQR, 49.99–80.2). The median BMI *z*-score of the NG tube feeding group was −1.13 (IQR, −3.2 to 0.28), that of the BMI percentile was 30.93 (IQR, 11.55–47.17), and that of the PIBW was 91.6 (IQR, 78.01–91.6) (Appendix A).

The baseline median CIR of the oral feeding group was 43.55 (IQR, 38.75–49.56), while that of the NG tube feeding group was 62.5 (IQR, 34.78–92.32). The baseline median PIR of the oral feeding group was 50.12 (IQR, 38.34–56.28), while that of the NG tube feeding group was 55.68 (IQR, 29.7–88.79). There were no significant differences in baseline nutrition between the two groups.

During the study period, the CIR and PIR of both groups improved (Figure 2, Appendix A). The oral feeding group showed greater improvement during the first six months (95% CI: 25.58–53.92, *p* = 0.01 for CIR; 95% CI: 15.67–70.01, *p* = 0.02 for PIR). The increase in nutritional intake in the oral feeding group was better than that in the NG tube feeding group (95% CI: −22.79–47.39, *p* = 0.01 for CIR; 95% CI: −3.95–49.89) (Figure 2, Appendix A).

### 3.5. Anthropometric Parameters and Nutritional Intake by Patient Muscle Tone

We compared the parameters according to the patients’ muscle tonicity (hypertonic vs. hypotonic). Figure 3 compares the anthropometric changes and nutritional intake changes between groups; detailed data are shown in Appendix A. In the hypertonic group, the PIBW (during the first six months: 95% CI: 2.14–16.42, *p* = 0.02; during the study period: 95% CI: −0.18–24.45, *p* = 0.05) and PIR (during the first six months: 95% CI: 12.33–56.00, *p* = 0.01; during the study period: 95% CI: 17.13–57.83, *p* = 0.01) were continuously improved during the study period. The BMI-for-age *z*-score of the hypertonic type group was also increased at the 6-month visit (95% CI: 0.16–1.69, *p* = 0.03). There were statistically significant increases in BMI-for-age percentile (95% CI: −5.01–23.67, *p* = 0.03), height-for-age *z*-score (95% CI: −1.76–0.2, *p* = 0.05), and CIR between the 6-month visit and the last visit (95% CI: 10.42–57.14, *p* = 0.03). In the hypotonic group, all anthropometric parameters, except height-for-age percentile and height-for-age *z*-score, and nutritional data improved without statistical significance.

## 4. Discussion

Nutritional support, including enteral feeding, helps solve the nutritional deficiencies of neurologically impaired children. The body composition parameters, including anthropometric variables, muscle mass, bone density, and fat deposition of these patients are reduced [24,25,26]; thus, special growth curves have been developed for patients with CP. Several studies have attempted to design growth charts for pediatric patients with CP at each GMFCS level [16,27,28]. However, these growth curves did not include children under two years of age. The ESPGHAN guideline recommends using the WHO growth chart in the anthropometric evaluation of children with neurological disabilities [29]. The 2017 Korean National Growth Charts were developed for normal and healthy children without any disabilities [15]. The ESPGHAN does not recommend that growth charts for patients with CP at each GMFCS level be used in the anthropometric and nutritional evaluation of pediatric neurologically impaired patients. However, there are no special growth curves for children with neurological diseases, and the growth of bedridden children is usually far from that of healthy children. Thus, along with applying the two standard growth curves for healthy children, we used the GMFCS curve to evaluate the degree of individual anthropometric and nutritional improvement in these patients.

IBW or optimal bodyweight [30] can be used to evaluate the degree of malnutrition. Waterlow defined malnutrition in pediatric patients using the PIBW calculation ((actual bodyweight/IBW) × 100) [20]. There are several methods for calculating IBW, and the most common methods used in children are the McLaren method [31], the Moore method [17], and the BMI method [32]. We chose the Moore method, using the same weight and height percentiles. We calculated IBW and BMI with estimated height using ulnar length and used these anthropometric parameters to evaluate the degree of catch-up growth; this estimated height would likely magnify the possible error, due to the possible gap between actual height and calculated height.

A previous study reported that the anthropometric parameters of children with CP aged 5 months to 17 years were improved 6 and 12 months after PEG tube insertion without serious complications [33], while another study reported similar outcomes [34]. Most other previous studies focused on complications [11,35] or comparisons between PEG tube insertion techniques [10,36,37,38,39,40].

To the best of our knowledge, this is the first study with a follow-up period of longer than 12 months (6.5-year total study period) to evaluate the anthropometric and nutritional effects after PEG tube insertion. We found that children achieved most of the catch-up growth and nutritional correction in the six months following PEG tube insertion. Weight-for-age, BMI-for-age *z*-score, and PIBW were improved, and the PIBW reached almost 100% at the 6-month follow-up visit. However, the changes in anthropometric parameters were minimal after six months. This finding implies that most of the catch-up growth was achieved in the first six months; thus, we should focus on nutritional care during this period since it appears to be important for the overall outcomes. One child became obese after the intervention period; it can be assumed that this is because individualized nutrition guidelines for different disease entities have not yet been developed. Therefore, continuous nutritional support is important to maintain the patients’ nutritional status.

The baseline BMI-for-age percentile, *z*-score, and PIBW of the NG tube feeding group were better than those of the oral feeding group. The oral feeding group increased caloric intake during the first six months after PEG tube insertion. Nutritional intake of the NG tube feeding group was increased without statistical significance; thus, we assume that children with NG tube feeding would not be affected by switching to a PEG tube. This result implies that PEG tube feeding is superior to NG tube feeding to provide stable nutritional support. Moreover, NG tube feeding can cause complications, such as gastrointestinal bleeding and aspiration. Indeed, seven of our 10 patients with NG tubes experienced bleeding ulcers, while the remaining three had aspiration problems prior to PEG insertion. We can assume that these complications prevented their caregivers from actively feeding the patients.

An Asian cultural background makes parents reluctant to subject their children to invasive procedures, such as NG tube placement, and a considerable number of parents in our study had refused PEG tube insertion for a long time. This repulsive emotion has a tendency to worsen malnutrition and delay the treatment of the underlying disease. In our study, two children under 24 months were maintained a normal or overweight status; this may be due to the short duration of diseases and earlier intervention and suggests that earlier nutritional intervention can prevent malnutrition. Considering previous studies demonstrating the improved quality of life of caregivers [41,42,43] and better prognosis with earlier interventions [34], clinicians should encourage caregivers to consider allowing PEG tube insertion earlier to better support their children’s health.

Another interesting result was that the BMI for-age *z*-score and PIBW in the hypertonic group improved significantly after PEG tube insertion, but did not improve in the hypotonic group despite similar increases in nutritional intake. Since the number of hypotonic children was small, we could not demonstrate statistical significance. To the best of our knowledge, this is the first study to show different nutritional improvement according to the muscle tonicity after PEG tube feeding. A previous study on hemiplegic CP of asymmetric growth between the affected and unaffected sides [44] reported that CP patients have different growth patterns and require different nutritional support depending on the CP type. Another study revealed that hypertonic children had lower fat mass, fat free mass, and total bodyweight [45]. Hypertonic children often experience swallowing difficulty, due to the increased muscle tone [46], making feeding difficult in these children. Thus, we assume that the actual effective nutritional volume is smaller than the total feeding volume. The actual effective nutritional volume may have been increased after PEG tube insertion; this seems to be an important reason for the anthropometric and nutritional improvement observed in hypertonic children. However, an extended study with a larger study population is warranted to fully validate this finding.

In comparison with that after the nutritional evaluation using growth curves for CP, the poor nutritional status was overly exaggerated when our patients were assessed against the 2017 Korea National Growth Charts [15,16], which explains the universal validation of CP growth curves. Moreover, there are no standardized growth curves or equations of nutrition tailored to the degree of rigidity.

The laboratory findings did not provide significant nutritional information during the study period, which demonstrated that the assessment of macronutrients using laboratory tests would not support the nutritional status of children with neurologic impairments. The ESPGHAN recommends micronutrient assessment (calcium; iron; zinc; vitamins C, D, and E; and selenium) [29]. Vitamin D is important in bone metabolism, and its deficiency is a high risk of fractures [25], while zinc is reported to be important for the prevention of seizures [47]. Children with epilepsy have been reported to have lower levels of zinc [48], while children with neurological impairments have lower levels of vitamin D [49]. Since these micronutrients are crucial for the good prognosis of patients, it is important to regularly test micronutrients and give the patients a sufficient dietary intake. In this study, vitamin D was checked routinely according to the recommendation of the PNST; the vitamin D levels were not particularly low and did not change significantly during the study. There were insufficient data on zinc levels to analyze. The provision of a sufficient amount of micronutrients according to the micronutrient level in regular follow-up will help to improve the nutrition of these children.

This study had several limitations. It was a retrospective study with a small number of patients. In the future, a large-scale study, including more detailed micronutrient data (such as vitamin C, zinc, and iron, and anthropometric parameters, e.g., fat mass, fat free mass, and bone mineral density), are needed to fully validate our findings.

## 5. Conclusions

When a child’s nutritional intake is insufficient, PEG tube insertion should be actively encouraged to ensure adequate nutritional support, even if he or she can eat orally. This must be explained to caregivers and physicians alike. Regular intervention by the PNST is important to ensure adequate nutritional support for children. In the future, the development of a growth chart for neurologically impaired children according to the degree of rigidity is required to ensure adequate nutritional support for this population. The findings of the current study might contribute to establishing individualized nutritional plans for children with neurological disabilities.

## Figures and Tables

**Figure 1 jcm-09-03295-f001:**
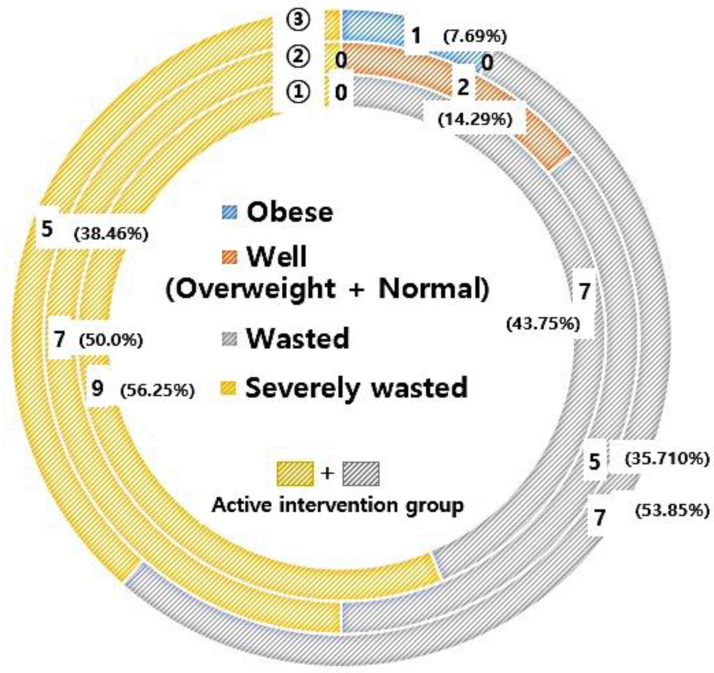
Nutritional evaluation during the study period according to body mass index-for-age *z*-score. (1) baseline visit; (2) 6-month visit; (3) Last visit.

**Figure 2 jcm-09-03295-f002:**
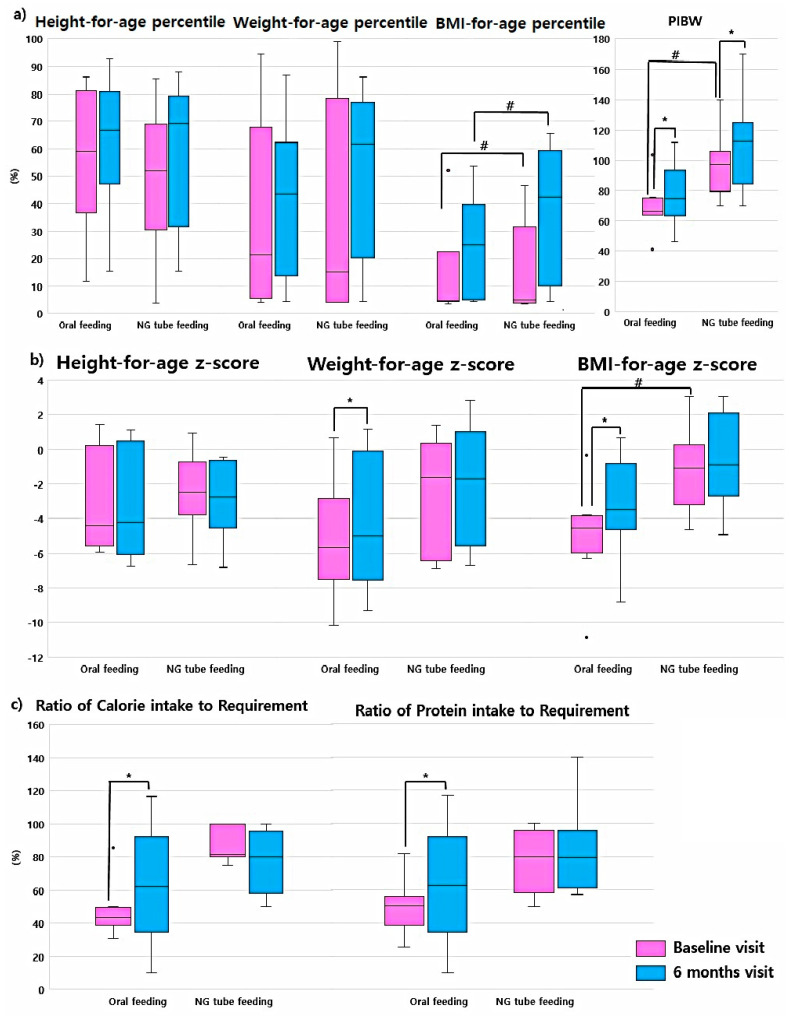
Comparisons of changes in anthropometric parameters and the ratio of nutritional intake between previous oral feeding group and NG tube feeding group. (**a**) Anthropometric parameters for-age percentile and percent ideal body weight. The percentile of each anthropometric parameter was calculated using the growth curve for cerebral palsy patients by Brooks et al. [16] and percent ideal body weight was calculated using the Moore method. (**b**) Anthropometric parameter for-age *z*-score. (**c**) Ratio of caloric intake to required amount and ratio of protein intake to required amount. NG tube feeding, fed via nasogastric tube before PEG tube insertion; oral feeding, fed orally before PEG tube insertion; PIBW, percent ideal body weight. * *p* < 0.05 by Wilcoxon signed-rank test; # *p* < 0.05 by Mann-Whitney *U* test.

**Figure 3 jcm-09-03295-f003:**
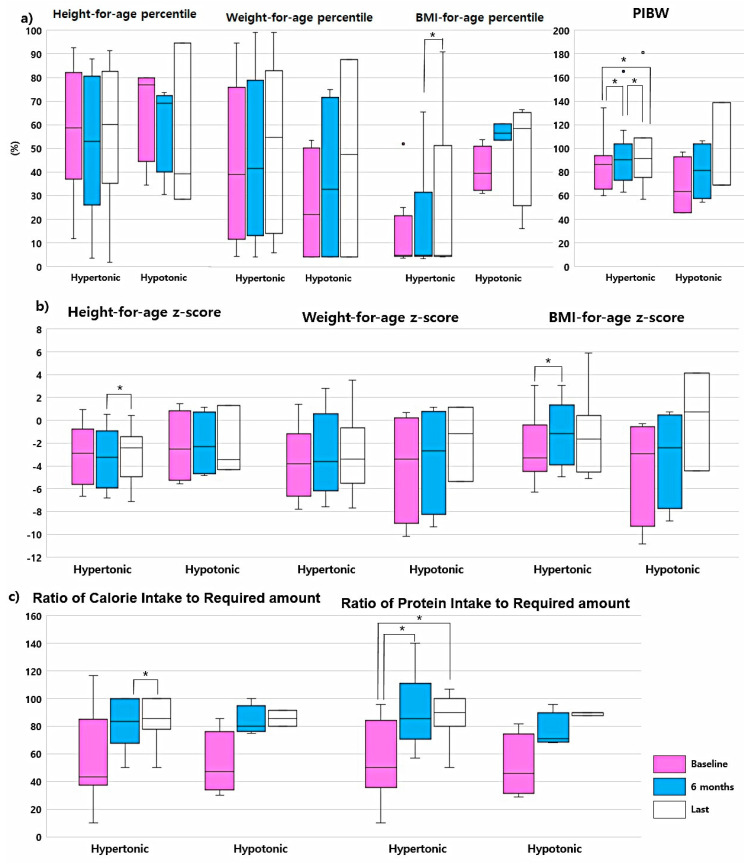
Comparisons of changes in anthropometric parameters and nutritional ratio between hypertonic and hypotonic groups. (**a**) Anthropometric parameters for-age percentile and percent ideal body weight. The percentile of each anthropometric parameter was calculated using the growth curve for cerebral palsy patients by Brooks and percent ideal body weight was calculated using Moore methods. (**b**) Anthropometric parameters for-age *z*-score. (**c**) Ratio of caloric intake to required amount and ratio of protein intake to required amount. NG tube feeding, fed via nasogastric tube before PEG tube insertion; oral feeding, fed orally before PEG tube insertion; PEG, percutaneous endoscopic gastrostomy; PIBW, percent ideal body weight. * *p*-value < 0.05 by Wilcoxon signed-rank test.

**Table 1 jcm-09-03295-t001:** Characteristics of the enrolled patients.

Characteristics	Number of Patients
Male/female, *n*	12/6
Median age (months) (IQR)	132.4 (43.0–180.7)
Study duration of intermedius (months) (IQR)	50.4 (33.9–71.8)
Previous feeding	
Oral feeding, *n*	8
NG tube feeding, *n*	10
Indications for PEG tube insertion	
Aspiration (oral/NG tube), *n*	8 (5/3)
Delayed oral phase (oral/NG tube), *n*	3 (3/0)
GI bleeding (oral/NG tube), *n*	7 (7/0)

**Table 2 jcm-09-03295-t002:** Neurological characteristics, according to disease entities of the enrolled patients.

Disease Entities	Number of Patients
Spasticity	14
Hypoxic ischemic encephalopathy	2
Cerebral palsy	4
Schizencephaly	1
Alpert syndrome	1
Lennox-Gastaut syndrome	3
Devic syndrome	2
Lissencephaly	1
Hypotonicity	4
Cerebral palsy	1
Rett syndrome	1
Lennox-Gastaut syndrome	1
Spinal muscular dystrophy	1

**Table 3 jcm-09-03295-t003:** Changes in the anthropometric parameters and nutrition during the study period.

Measurement	Median (IQR)			*p*-Values		
	Baseline Visit	6-Month Visit	Last Visit	*	**	***
Height percentile ^(a)^	59.3 (41.04–80.47)	55.06 (33.7–76]	53.48 (33.6–83.69)	0.061	0.308	0.3
Weight percentile ^(a)^	37.83 (9.96–62.23)	41.39 (11.87–77.5]	51.15 (13.91–83.05)	0.134	0.859	0.331
BMI percentile ^(a)^	11.64 (4.55–35.34)	26.03 (4.55–54.23]	35.93 (4.54–58.64)	0.158	0.009	0.136
PIBW ^(a)^	81.41 (63.91–93.79)	90.44 (68.68–103.77)	90.41 (69.09–108.68)	0.003	0.041	0.056
Height *z*-score ^(b)^	−2.88 (−5.57–(−0.73))	−3.25 (−5.35–(−0.55))	−2.62 (−4.52–(−1.3))	0.255	0.034	0.158
Weight *z*-score ^(b)^	−3.85 (−6.67–(−0.84))	−3.61 (−6.21–0.6)	−2.77 (−5.41–(−0.42))	0.011	0.136	0.975
BMI *z*-score ^(b)^	−3.29 (−4.65–(−0.55))	−1.18 (−4.41–0.46)	−1.61 (−4.48–0.72)	0.005	1	0.158
CIR	45.42 (37.22–83.93)	80 (75–100)	85.72 (79.45–95.32)	0.015	0.953	0.016
PIR	50.12 (35.84–82.2)	84 (69–101.34)	90(82.5–95.91)	0.004	1	0.002

*p*-values were determined using the Wilcoxon signed-rank test. * *p*-value for differences between the initial visit and the 6-month visit. ** *p*-value for differences between the 6-month visit and the last visit. *** *p*-value for differences between the initial visit and the last visit. BMI: Body mass index, CIR: Calorie intake compared to the recommended requirement, IQR: Interquartile range, PIBW: Percent of ideal bodyweight, PIR: Protein intake compared to the recommended requirement. ^(a)^ Percentile measured based on the growth curves for cerebral palsy according to sex and the Gross Motor Function Classification System level. ^(b)^
*z*-score calculated based on the 2017 Korea National Growth Chart data.

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
