# Peer review of "Percutaneous Endoscopic Gastrostomy and Nutritional Interventions by the Pediatric Nutritional Support Team Improve the Nutritional Status of Neurologically Impaired Children"

_jcm, 2020, doi:10.3390/jcm9103295_

Round 1

Reviewer 1 Report

The comments are in the attached file, because I cannot paste them.

Reviewer 2 Report

In the description of the patients (figure 1) I would do better the two separate tables that explain the case series, one with the nutritional characteristics and one with the neurological characteristics of the diseases.

Author Response

In the description of the patients (figure 1) I would do better the two separate tables that explain the case series, one with the nutritional characteristics and one with the neurological characteristics of the diseases.

  • Thank you for your kind opinion. We modified figure 1 to table 1 and table 2 according to your advice.

We all appreciate your valuable comments. Your comments make our manuscript more precise.

We look forward to your positive answer.

Round 2

Reviewer 1 Report

I think that the quality of the document has improved after the changes made by the authors. However, regarding the changes in anthropometric data the results presented (lines 192-193) are for 16 children (100%), when they previously say that they included 18. It must be clarified.

Author Response

I appreciate your precise comment. We are sorry for the lack of explanation. Out of total 18 children, 16 children were aged ≥ 24 months, and these 16 refer to children who can be evaluated by BMI-for-age according to WHO growth standard and Korea National Growth Chart guidelines. Other 2 children were under 24 months and could not be evaluated by BMI-for-age. Instead, they were assessed by weight-for-height curve according to these guidelines. In addition, we have analyzed weight-for-height z-score data along with the BMI-for-age z-score data, and the difference between the initial visit and the 6-month visit was also significant (P=0.003). However, we have excluded this result from the manuscript and described the data separately, because BMI-for-age and weight-for-height were different parameters. We added information in the method (Line 102-104, and 119-121) and result (Line 141 and 161-166) session. And we added a discussion of these 2 children under 24 months (Line 275-276). Thank you for the comment again.